# Epidemiology and Surgical Management of Foreign Bodies in the Liver in the Pediatric Population: A Systematic Review of the Literature

**DOI:** 10.3390/children9020120

**Published:** 2022-01-18

**Authors:** Francesca Gigola, Chiara Grimaldi, Kejd Bici, Marco Ghionzoli, Claudio Spinelli, Paolo Muiesan, Antonino Morabito

**Affiliations:** 1Department of Pediatric Surgery, Meyer Children’s Hospital, 50139 Florence, Italy; francesca.gigola@unifi.it (F.G.); kejd.bici@meyer.it (K.B.); marco.ghionzoli@meyer.it (M.G.); antonino.morabito@unifi.it (A.M.); 2Department of Surgical, Medical, Molecular Pathology and of the Critical Area, University of Pisa, 56100 Pisa, Italy; claudio.spinelli@unipi.it; 3Department of Hepatobiliary Surgery, Careggi University Hospital, 50134 Florence, Italy; paolo.muiesan@unifi.it

**Keywords:** liver, foreign body, children, penetrating, trauma

## Abstract

Retention of foreign bodies (FB) in the liver parenchyma is a rare event in children but it can bring a heavy burden in terms of immediate and long-term complications. Multiple materials can migrate inside the liver. Clinical manifestations may vary, depending on the nature of the foreign body, its route of penetration and timing after the initial event. Moreover, the location of the FB inside the liver parenchyma may pose specific issues related to the possible complications of a challenging surgical extraction. Different clinical settings and the need for highly specialized surgical skills may influence the overall management of these children. Given the rarity of this event, a systematic review of the literature on this topic was conducted and confirmed the pivotal role of surgery in the pediatric population.

## 1. Introduction

Retention of a foreign body (FB) in the liver is a rare circumstance, however it can lead to a heavy burden in terms of immediate and long-term complications. A FB can reach the liver through three different routes: direct penetrating injury, via the gut following an ingestion, or through the bloodstream [1,2]. Multiple objects can be retrieved from the liver, such as sewing needles, hair pins or military equipment such as pellets or gun shots. In addition, some FBs may result from medical actions: insertion of a FB due to inaccurate surgical procedures or migration of medical devices [3,4,5].

Moreover, clinical signs may differ based on the type of FB, on the timing after initial injury, and on the way of entry. Hence, management of liver FBs may greatly differ.

Patients are often completely asymptomatic but the persistence of a foreign material inside the parenchyma may cause severe complications, usually infections such as liver abscess, hepatic granuloma, pseudotumor [2,6,7,8] or dislocate even over the long-term, possibly causing biliary or vascular damage [9,10,11].

Given the rarity of these events, management may be heterogeneous. Some authors advocate for conservative treatment in case of asymptomatic FBs based on the absence of clinical manifestation even after a long follow-up, while others support an operative approach to prevent the risk of vascular damage [1,2].

To evaluate the general management of this rare entity and determine whether surgery should be recommended as the standard approach, a systematic review of the literature was conducted.

## 2. Material and Methods

We conducted a systematic review to identify the most relevant studies focused on hepatic FBs in children. The study strategy complied with the Preferred Reporting Items for Systematic Reviews and Meta-Analysis (PRISMA) guidelines. The present review was not registered on the PROSPERO database. The literature search was conducted on MEDLINE-PubMed and EMBASE using the following terms: liver, trauma, penetrating, children, weapon, and foreign object and foreign body. Publications between 1 January 2000 and 14 August 2021 were considered. All case reports included in this review were analyzed using the JBI Critical Appraisal Checklist for Case Reports [12]. The only retrospective study included was assessed using an adapted version of the Appraisal tool for Cross-Sectional Studies [13].

Inclusion criteria were age < 18 years and FB retention in the liver. Manuscripts reporting on extra-hepatic foreign bodies, studies that reported non-pediatric cases and that did not have access to a full article in English were excluded.

After the exclusion of duplicates, articles were first screened based on their title and abstract, and those considered eligible for inclusion were read in full copy. Three reviewers (C.G., F.G., K.B.) screened the identified studies independently and extracted the data. Any case of disagreement was resolved by consensus. The reference lists of all eligible papers were inspected to find other additional articles discussing the same topic and not found through the initial search.

Data extraction was carried out using a spreadsheet including, if available, the following data: year of publication, study design, number of patients, type of foreign body, clinical presentation, type and timing of surgery and outcome.

Given the paucity of patients identified within the selection criteria, the results are reported as a narrative review.

## 3. Results

After removing the duplicates, the literature search produced a total of 321 articles. Two additional articles were found in the reference lists and added, for a total of 323 manuscripts. A total of 249 papers were excluded based on the title, in addition 41 were excluded following screening of the abstract. Out of the remaining 33 studies, 16 were excluded because they did not include pediatric cases and FB retention, or the full text was either not available or not in English. In addition, one article was excluded because of an overlap between two manuscripts, both referring to the same patient.

The review screening process is detailed in Figure 1.

The articles selected for the review were published between 2003 and 2020 and involved a total of 16 patients with hepatic FB retention. The median age at diagnosis was five years (range: 3 months–16 years). There were six females and eight males. Gender was not specified in two cases. A complete list of articles included in the review and the data extracted is presented in Table 1.

The FBs retained were: sewing needle (9) [2,14,15,16,17,18,19,20], medical device (3) [4,5,21], gun pellet (2) [22,23], and pin (2) [24,25] The medical devices consisted in the distal part of a ventricular-peritoneal shunt [4], a gastrostomy bumper that was incorrectly inserted through the abdominal wall via the anterior surface of the left liver lobe [5] and a Kirshner wire that migrated to the liver from the hip [21].

The route of access to the liver parenchyma was detailed in seven cases: three patients ingested the FB [15,24,26], in four cases a direct penetration was reported (two gunshots [22,23], one ventricular-peritoneal shunt [4] and one gastrostomy bumper [5]), while in nine cases the route was either uncertain or not described [2,14,16,17,18,19,20,21,25].

In nine patients the precise medical history (mode of entry, onset of symptoms, time of persistence of the FB in the liver) could not be collected or was unspecific, mostly in younger children with ingestion of a small FB [2,14,16,17,18,19,20,25,26].

Interestingly 43.8% of patients (seven out of 16) were asymptomatic at diagnosis and the FB was detected incidentally. Two sewing needles, one pin, one dislocated Kirschner wire were detected upon follow-up X-rays [2,14,21,24]. In two cases the FB was diagnosed upon imaging for appendicitis and laryngitis [17,26].

The most frequent clinical signs were abdominal pain (five patients) [4,15,16,17,25] and vomiting (four patients) [4,16,17,26]. In five cases laboratory findings showed increased white cell blood count [4,15,16,18,25] and in three cases increased hepatic enzymes [16,18,25].

More specific symptoms such as gastrostomy malfunction and neurological impairment were detected in patients with a retained medical device (gastrostomy malfunction and ventricular-peritoneal shunt malfunction, respectively) [4,5].

In most cases (81.2%) a plain X-ray led to the diagnosis of metallic FBs [2,14,15,16,17,18,19,20,21,23,24,25,26] while 56% of patients (nine out of 16) required an abdominal ultrasound [2,16,17,19,20,21,24,25,26]. A CT-scan was performed in 43.7% of cases (seven out of 16) [2,4,16,17,18,19,20] as preoperative imaging. Finally, two patients underwent gastroscopy [5,15] while for one patient no preoperative data are available [22].

Although the duration of symptoms before hospital admission was unclear in most cases, the mean period of retention of the FB was 5.8 months (range: 0–3 years). Most FBs were in the right hepatic lobe (nine out of 16) [4,15,16,17,18,19,21,23,24], in three cases the FBs were in the left hepatic lobe [5,14,25] one in the quadrate lobe [20], one in the caudate lobe [2] and two studies did not specify the position [22,26].

Overall, 12 patients (75%) underwent surgical removal of the FB, mostly by laparotomy (10 patients) [5,14,15,16,17,18,19,20,21,23] while a laparoscopic approach was preferred in two cases [2,26]. Finally, in one case the FB was endoscopically removed [4,24]. One patient spontaneously delivered the FB (ventriculo-peritoneal shunt) with stools. All patients had a favorable follow-up without any surgical complication reported. Two papers focusing on imaging did not provide details on surgical management [22,25].

An overview of patient demographics and clinical presentation, FB characteristics and management is reported in Table 2.

## 4. Discussion

As reported in multiple adult and pediatric series and case reports, retained hepatic FBs are distinguished in three categories, based on the route used to reach the liver: penetrating, ingested, and bloodstream [2,15].

Medical FBs may, as well, migrate into the liver and consist in surgical objects such as clips, t-tubes, gauzes, or medical sutures which are retained in the liver parenchyma usually following surgical procedures [27,28,29,30,31].

Pediatric cases are exceedingly rare: our thorough search of the English literature found 15 manuscripts reporting single cases and one retrospective study for a total of 16 children.

Aims of this review were to define the clinical presentation and management of liver retained foreign bodies in children. Patients usually are younger children, less than three years old [1,2,15] ingesting sharp-edged objects such as fish bones, sharp bone pieces, cocktail sticks, and sewing needles [1,8,32,33]. A specific medical history with the intention to determine the timing and modality of ingestion is challenging for almost all patients. This difficulty is similarly reported even for older patients [34]. For those children having a detailed history related to the initial event, the median time at diagnosis is six months. Moreover, one-third of the patients are asymptomatic, and the FBs are incidentally detected, since most ingested FBs pass through the gastrointestinal tract uneventfully [2,14,17,21,26]. Bowel perforation rarely occurs following a swallowed FB, usually at the ileocecal and rectosigmoid regions in adults, or in the stomach and duodenum in children [8,15,35]. Swallowed FBs in cases as such, may migrate to the left liver, probably due to the proximity of duodenum and stomach to the left lobe [1]. Akçam et al. [24] and Azili et al. [15] described two patients with gut perforation: the first with a duodenum perforation and a pin retained in the right liver whilst the latter described a FB in the left hepatic lobe secondary to a spontaneously healed gastric perforation [15,24]. The event was described as subclinical or asymptomatic in both cases. A different subset of patients is represented by those suffering from migration of medical devices (e.g., gastrostomy or ventricular-peritoneal shunt) and penetrating foreign bodies such as bullets or pellets. In these patients, the right lobe of the liver is the most frequent localization, probably due to its greater surface area [4,5,22,23].

Pediatric gunshot injuries are a frequent cause of death in some countries [22,36,37]. Although penetrating abdominal injuries are usually considered surgical emergencies [38] multiple manuscripts have demonstrated the safety of a non-operative approach [39,40]. Absence of metallic debris is mandatory to attempt a management may be heterogeneous conservative management: imaging with CT scan is deemed useful to obtain both trajectory information and monitor selected patients [38]. Large series usually report data from the adult population; however, the recently published World Society of Emergency Surgery (WSES) Pediatric guidelines recommend a similar approach in children with penetrating injuries [41].

Regardless of the nature of the FB, radiological imaging is essential to tailor the operative management.

A plain abdominal X-ray usually detects a FB [15,16,17,18,19,20,24] however it is imprecise in defining the exact position and anatomical structures involved. Therefore, further investigations, such as ultrasound or CT scan, are usually needed. A CT scan is recommended as a routine examination before surgery [42,43]. In our review, seven out of 16 patients underwent a CT scan [2,4,16,17,18,19,20] preoperatively. When the FB is still partially retained in the gut, a gastroscopy might be proposed with both diagnostic and therapeutic intention [5,15,24]. A flowchart of patient management is summarized in Figure 2.

As for liver penetrating injuries, pellets may enter the abdominal wall and pass through the liver on their trajectory: in this setting the liver lesion can be treated conservatively following the usual guidelines for the treatment of parenchymal laceration in blunt abdominal trauma. In the rare cases when a metallic shard remains embedded in the liver parenchyma, treatment should change according to the following management of retained FBs [2,7].

Revision of all published data on surgical management of liver FBs confirm that surgical extraction needs to be considered in all patients, mostly based on lessons learned from the adult literature [44]. Published experiences in adults have reported miscellaneous long-term complications related to retained FBs in the liver: delayed surgery may lead to liver abscess, hepatic granuloma, pseudotumor [2,6,7,8] or dislocation possibly causing biliary or vascular damage [9,10,11].

In our review, three patients developed liver abscess due to the presence of a retained FB [16,22,25], in one case the FB migrated to the gallbladder [16] and in a second, after 14 months, the retained sewing needle migrated to the right kidney [26] In one patient the dislocation of the FB from the right liver caused a fistulation inside the transverse colon [4]. Interestingly, none of these patients developed complications following surgery.

Given the above, we assume that surgical extraction should always be proposed in highly skilled settings with an adequate ahead of schedule discussion.

## 5. Conclusions

This is, to our knowledge, the first systematic review on management of liver FBs in children.

Although this is a very rare event in the pediatric population, it can lead to serious clinical manifestations. Despite the paucity of cases reported in children, based on this extensive analysis of the literature, a surgical extraction of the retained FB appears to be feasible and safe, and should be recommended in order to minimize short- and long-term complications.

## Figures and Tables

**Figure 1 children-09-00120-f001:**
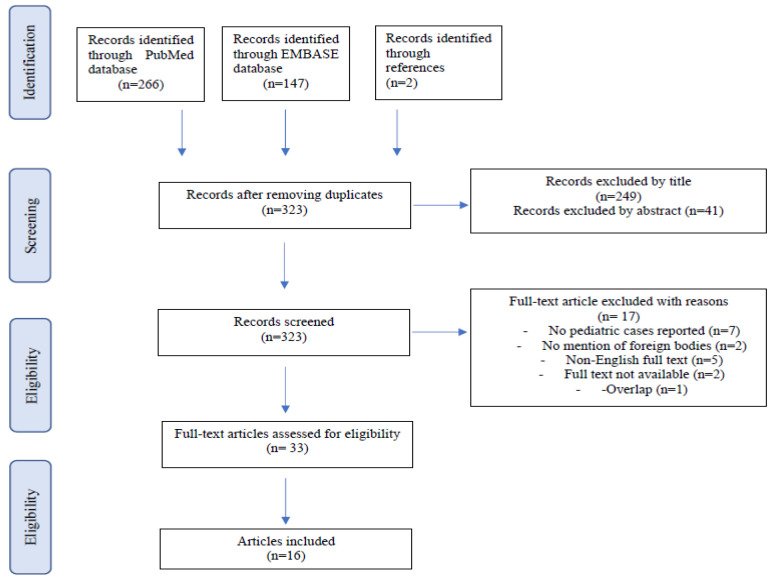
Review screening process.

**Figure 2 children-09-00120-f002:**
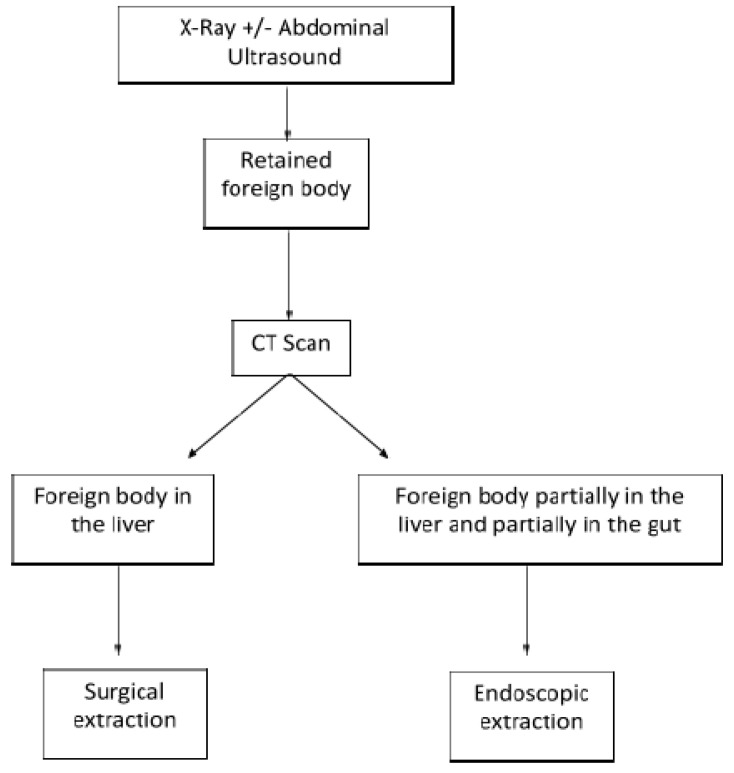
Flowchart for the management of pediatric liver foreign body.

**Table 1 children-09-00120-t001:** List of papers included in the review and findings.

Article	Year	Article Type	Number of Patients	Age	Sex	Foreign Body	Clinical Presentation	Imaging	Duration of Symptoms	Timing	Mode of Entry	Position in the Liver (Lobe)	Surgery	Surgical Complications	Outcome
Nishimoto Y. [14]	2003	Case report	1	1 y	M	Sewing needle	Asymptomatic	X-ray	Asymptomatic	Unknown	Unknown	Left	Laparotomy, extraction	None	Discharged on 8th post-op day
Thipphavong S. [4]	2004	Case report	1	12 y	F	Ventriculoperitoneal shunt	Vomiting and headache, abdominal pain and augmented white blood cell count	CT	5 d	4 m	Iatrogenic	Right lobe	Spontaneous passing through the anus	No surgery performed	Asymptomatic at 1-year follow-up
Meeks T. [23]	2004	Case report	1	22 m	F	Air-powdered pellet	Asymptomatic	X-ray, Angiography	Unknown	Immediately	Penetration	Right	Laparotomy	None	Discharged on 8th post-op day
Marya KM. [21]	2006	Case report	1	5 y	U	Kirschner wire	Asymptomatic	X-ray, Abdominal US	Unknown	4 w	Iatrogenic	Right	Laparotomy, extraction	None	Asymptomatic at 1-year follow-up
Azili MN. [15]	2007	Case report and review of literature	1	14 y	F	Sewing needle	Abdominal pain, fever, increased white blood count	X-ray, Gastroscopy	Unknown	1 m	Ingestion	Right	Laparotomy, extraction	None	Discharged on 7th post-op day
Akçam M. [24]	2009	Case report	1	5 y	M	Pin	Asymptomatic	X-ray, Abdominal US	Unknown	3 m	Ingestion	Unknown	Endoscopic removal	None	Discharged on 1st post-op day
Dominguez S. [2]	2009	Case report	1	3 y	M	Sewing needle	Asymptomatic	X-ray, Abdominal US, CT	Unknown	Unknown	Unknown	Caudate	Laparoscopy, extraction	None	Asymptomatic at 19-months follow-up
Avcu S. [16]	2009	Case report	1	16 y	F	Sewing needle	Abdominal pain, fever, nausea, vomiting, increased white blood count, increased AST, ALT, LDH, ALP.	X-ray, Abdominal US, CT	Unknown	Unknown	Unknown	Right	Laparotomy, extraction	None	Unknown
Bakal U. [17]	2012	Case reports	1	14 y	M	Sewing needle	Abdominal pain and vomiting (simultaneous appendicitis)	X-ray, Abdominal US, CT	1 d	Unknown	Unknown	Right	Laparotomy, extraction	None	Asymptomatic at 1-year follow-up
Xu BJ. [18]	2013	Case report	1	5 m	M	Sewing needle	Mild respiratory symptoms, increased white blood count, increased ALT, AST, bilirubin	X-ray, CT	3 d	Unknown	Unknown	Right	Laparotomy, extraction	None	Asymptomatic at 2-months follow-up
Adams S.D. [5]	2014	Case report	1	6 y	F	Gastrostomy bumper	Persistent discharge, trouble advancing and rotating the tube, “buried bumper” syndrome	Gastroscopy	18 m	3 y	Iatrogenic	Left	Laparotomy, extraction	None	Discharged on 7th post-op day
Hyak JM. [22]	2020	Retrospective study	1	U	U	Pellet fragment	11 cm liver abscess	Unknown	Unknown	1 m	Penetration	Unknown	Unknown	Unknown	Unknown
Le Mandat-Schultz A. [26]	2003	Case report	1	11 m	M	Sewing needle	Cough and vomiting (simultaneous laryngitis)	X-ray, Abdominal US	Unknown	Unknown	Ingestion	Unknown	Laparoscopy, extraction	None	Discharged on 2nd post-op day
Demir S. [19]	2020	Case report	1	11 y	M	Sewing needle	Left armpit and chest pain	X-ray, Abdominal Ultrasound, CT	Unknown	Unknown	Unknown	Right	Laparotomy, extraction	None	Asymptomatic at 6-months follow-up
Saitua F. [20]	2009	Case report	1	3 m	M	Sewing needle	Mild respiratory symptoms (cough and minor respiratory difficulty)	X-ray, Abdominal Ultrasound, CT	2 d	Unknown	Unknown	Quadrate lobe	Laparotomy and manual extraction	None	Discharged on 3rd post-op day
Ariyuca S. [25]	2009	Case report	1	16 y	F	Pin	Epigastric pain, abdominal tenderness, anorexia, elevated liver enzymes and white blood count	X-ray, Abdominal Ultrasound	Unknown	Unknown	Unknown	Left	Unknown	Unknownn	Unknown

Abbreviations: d: days; m: months; y: years; w: week.

**Table 2 children-09-00120-t002:** Patients demographics and clinical presentation.

	*n*	%
Gender		
Male	8	50%
Female	6	37%
Unknown	2	13%
**Type of foreign body**		
sewing needle	9	56%
pin	2	13%
gun pellet	2	13%
Medical devices	3	18%
**Route of access**		
Unknwon	9	56%
Ingestion	3	19%
Penetration	4	25%
**Clinical presentation**		
Asymptomatic	7	44%
Abdominal pain	5	31%
Vomiting	4	19%
Other	2	12%
**Imaging**		
X-ray	13	81%
Abdominal Ultrasound	9	56%
CT-scan	7	44%
Digestive endoscopy	2	13%
Angiography	1	6%
Unknown	1	6%
**Position in the liver**		
Right lobe	9	56%
Left lobe	3	19%
Quadrate lobe	1	6%
Caudate lobe	1	6%
Unknown	2	13%
**Intervention**		
Surgical removal	12	75%
Endoscopic removal	1	6%
Spontaneus delivery	1	6%
Unknown	2	13%

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
