# Peer review of "Epidemiology and Surgical Management of Foreign Bodies in the Liver in the Pediatric Population: A Systematic Review of the Literature"

_children, 2022, doi:10.3390/children9020120_

Round 1

Reviewer 1 Report

1. The number of articles included differ between the result section and figure 1. In the result section, line 78, the authors have written that “after removing the duplicates, the literature search produced a total of 321 articles” while in figure 1 they have mentioned that they have included 323 articles after removing duplicates. In reality, how many articles were included? Please clarify. 

2. In line 157, the authors have mentioned that a study has shown gut perforation related to foreign body ingestion. Since gut perforation could be a serious and life threatening condition, I am wondering if that study has discussed any complications associated with gut perforation?

3. Since the articles included in this study are heterogeneous in terms of the type of foreign bodies, its location in the human body and the time of its presence, it is difficult to understand if there are any specific symptoms or complications associated with particular type of foreign bodies, its location and duration of stay in the human body.

4. As the literature on the retention of foreign bodies in the hepatic liver parenchyma of the pediatric population is relatively rare, the study is unable to comprehensively enlighten the current understanding of this condition. Therefore, the author may consider including more references on the effects and consequences of foreign body’s penetration in adults which can at least give some idea what can be expected in the pediatric population.      

5. Figure 1 is showing up some weird symbol throughout, remove it.  

Reviewer 2 Report

Comment:

I always expect a visionary flow chart that came out of a Systematic Review, which provides the reader a handy diagnostics and treatment options. However, I did not see it here with the current version of the manuscript. Manuscript ID: children-1494290 [Type of manuscript: Systematic Review Title: Epidemiology and surgical management of foreign bodies in the liver in the pediatric population: a systematic review of the literature Authors: Francesca Gigola, Chiara Grimaldi *, Kejd Bici, Marco Ghionzoli, Claudio Spinelli, Paolo Muiesan, Antonino Morabito] attempted to provide a systematic review of the literature. However, the authors did not convincingly analyze the literature and came out with any insight and vision. Below specific comments, not exhaustively listed, should be addressed for better clarity and logic flow.

Specific comments:

  • Abstract: “Multiple materials can migrate inside the liver via the intestine after ingestion, via the bloodstream, directly after penetration through the abdominal wall or after surgical procedures.
  • Rather than a general introduction in Abstract, a Smash-and-grabs-like conclusion should be given on “a systematic review of the literature on this topic,” in particular, “Given the rarity of this event.”
  • Too vague to make any sense in the Introduction of the current version. E.g., a foreign body (FB) in the liver could be harmless. The authors need to get down some specific FB to illustrate their burden point. Given examples (fishbone, gauge) on Lines 36 – 41, the term “foreign object” (FO) is preferred over the term foreign body (FB).
  • All of these citations should spread out with specific FB or FO, including “liver abscess, hepatic granuloma, pseudotumor [2,6–8] or dislocate even over the long-term, 36 possibly causing biliary or vascular damage [9–11].”
  • Table 1 should be with a table title. The table should be provided with a column of post-surgery outcomes.
  • Page 5, the flow chart diagram should provide a Figure legend and description. Lines 40-41 were for 64-years-old. In this flow chart, they filtered out non-pediatric patients: Why did they cite a 64-years-old case?
  • Line 42: “clinical and surgical management” – Yet. Table 1 was on surgery only. What clinical management?
  • Line 43: what was their “conservative treatment?” Reasoning? Clinical outcomes?
  • Line 44: What was their “more aggressive approach?” reasoning? Clinical outcomes?
  • Line 71: “using an internally piloted spreadsheet including” – What did they mean internally? Their own data? If Not as cited in Table 1, delete “internally.”
  • Line 75: “as a narrative review and presented in tables” – Tables? Where?
  • Lines 97-129 statements should be presented with a Table for better summary views, complimentary to Table 1 that reports one case only.
  • Lines 190-191: “in the presence of symptoms (abdominal pain or respiratory distress) [15–20,24–26]” – outrageous nine citations are clustered on one spot, which does not make any sense to the reader.
  • Lines 130-224: the authors regurgitated some facts in literature – somewhat exhaustively listed - without any systematic analyses to extract their themes (Not a Discussion).
  • Lines 225-234: Conclusion taught the reader nothing new, no insight – Why a reader needs to read their review?
  • I always expect a visionary flow chart that came out of a Systematic Review, which provides the reader a handy diagnostics and treatment options. However, I did not see it here with the current version of the manuscript.

Round 2

Reviewer 1 Report

The authors have revised the manuscript as per the comments

Reviewer 2 Report

the revision has fully addressed my comments.